# Immunohistochemical Stain-Aided Annotation Accelerates Machine Learning and Deep Learning Model Development in the Pathologic Diagnosis of Nasopharyngeal Carcinoma

**DOI:** 10.3390/diagnostics13243685

**Published:** 2023-12-18

**Authors:** Tai-Pei Lin, Chiou-Ying Yang, Ko-Jiunn Liu, Meng-Yuan Huang, Yen-Lin Chen

**Affiliations:** 1Department of Life Sciences, National Chung Hsing University, Taichung 402, Taiwan; fionlin0211@gmail.com; 2Institute of Molecular Biology, National Chung Hsing University, Taichung 402, Taiwan; cyyang@dragon.nchu.edu.tw; 3National Institute of Cancer Research, National Health Research Institutes, Tainan 704, Taiwan; kojiunn@nhri.edu.tw; 4Graduate Institute of Medicine, College of Medicine, Kaohsiung Medical University, Kaohsiung 807, Taiwan; 5Institute of Clinical Pharmacy and Pharmaceutical Sciences and Institute of Clinical Medicine, National Cheng Kung University, Tainan 701, Taiwan; 6Department of Pathology, Tri-Service General Hospital, National Defense Medical Center, Taipei 114, Taiwan

**Keywords:** immunohistochemistry, nasopharyngeal carcinoma, diagnose, machine learning

## Abstract

Nasopharyngeal carcinoma (NPC) is an epithelial cancer originating in the nasopharynx epithelium. Nevertheless, annotating pathology slides remains a bottleneck in the development of AI-driven pathology models and applications. In the present study, we aim to demonstrate the feasibility of using immunohistochemistry (IHC) for annotation by non-pathologists and to develop an efficient model for distinguishing NPC without the time-consuming involvement of pathologists. For this study, we gathered NPC slides from 251 different patients, comprising hematoxylin and eosin (H&E) slides, pan-cytokeratin (Pan-CK) IHC slides, and Epstein–Barr virus-encoded small RNA (EBER) slides. The annotation of NPC regions in the H&E slides was carried out by a non-pathologist trainee who had access to corresponding Pan-CK IHC slides, both with and without EBER slides. The training process utilized ResNeXt, a deep neural network featuring a residual and inception architecture. In the validation set, NPC exhibited an AUC of 0.896, with a sensitivity of 0.919 and a specificity of 0.878. This study represents a significant breakthrough: the successful application of deep convolutional neural networks to identify NPC without the need for expert pathologist annotations. Our results underscore the potential of laboratory techniques to substantially reduce the workload of pathologists.

## 1. Introduction

Nasopharyngeal carcinoma (NPC), an epithelial cancer originating in the nasopharynx epithelium [1], displays a unique geographic prevalence compared to other malignancies. According to data sourced from the International Agency for Research on Cancer, in 2018, approximately 129,000 individuals received NPC diagnoses, accounting for only 0.7% of all tumor cases [2,3]. Notably, NPC exhibits a significantly high incidence in East and Southeast Asia, including Taiwan [4]. The World Health Organization (WHO) classifies NPC into three histological subtypes: keratinizing squamous cell carcinoma, non-keratinizing (differentiated or undifferentiated) carcinoma, and basaloid carcinoma. In regions with high incidence, undifferentiated carcinoma prevails, constituting over 95% of NPC cases and being associated with a more favorable survival prognosis [1,4]. The development of NPC is believed to result from complex interactions among factors such as Epstein–Barr virus (EBV) infection, genetic predisposition, and environmental influences like alcohol consumption and tobacco use [5]. The management of NPC has significantly improved thanks to advancements in radiotherapy technology, the utilization of induction and concurrent chemotherapy, and the implementation of precise cancer staging systems [1,4]. Management aligns with the consensus of the American Joint Committee on Cancer (AJCC) tumor–node–metastasis (TNM) classification system for NPC [6]. While pathological diagnosis is usually straightforward in daily practice for pathologists, some cases with few carcinoma cells within the biopsy slides may occasionally be missed. Hence, there is an urgent need for a second evaluation of these high-risk biopsies. Double confirming the malignancy diagnosis by another pathologist is a common practice in many pathology laboratories to maintain high-quality control. However, such double confirmation mechanisms are lacking for these high-risk slides, which are sometimes erroneously diagnosed as benign by only one pathologist.

The advancement of artificial intelligence (AI) in the field of pathology has evolved hand in hand with the growth of digital pathology. The roots of this evolution can be traced back to the 1960s, when Prewitt et al. [7] initiated a pivotal process. They began by scanning basic images derived from microscopic fields, typically found in blood smears. These datasets were intricately linked with clinical outcomes and genomic data, fostering a fertile ground for significant advancements in AI research within the realms of digital pathology and oncology [8,9].

The evolution of AI models in the field of pathology has followed a fascinating trajectory that has significantly transformed the landscape of medical diagnostics. This journey can be delineated from expert systems to traditional machine learning (ML), and eventually, to deep learning (DL), reflecting the relentless pursuit of more accurate and efficient diagnostic tools.

The advent of deep learning (DL) revolutionized the field of pathology by enabling AI systems to directly learn from raw data, thereby bypassing the laborious feature engineering process [10]. Deep learning models, particularly neural networks with multiple hidden layers, have the capacity to automatically extract intricate and subtle patterns within medical images and data. Convolutional neural networks (CNNs), fully convolutional networks (FCNs), recurrent neural networks (RNNs), and generative adversarial networks (GANs) have become pivotal tools in this era, driving extensive research into DL-based AI applications in pathology. These algorithms have harnessed the surge in computational processing power and have allowed for the development of sophisticated models capable of detecting anomalies, segmenting tumors, and even generating synthetic medical images with remarkable accuracy.

The primary objective behind integrating AI into pathology is to address the inherent limitations of subjective visual assessments by human pathologists [11,12]. While pathologists possess remarkable expertise, the variability in interpretations and the potential for fatigue-induced errors have prompted the exploration of AI as a complementary diagnostic tool. AI systems excel at processing vast amounts of data rapidly and consistently, leading to more precise and reliable measurements, ultimately aiming to enhance the quality and efficiency of tumor treatment [13].

The ongoing evolution of AI models in pathology represents a powerful synergy between human expertise and machine intelligence, promising to augment the capabilities of medical professionals while ensuring that patients receive the best possible care. It is a field characterized by relentless innovation, with a commitment to improving diagnostic accuracy, reducing errors, and advancing the frontiers of medical knowledge.

Critical to the role of pathologists is the accurate differentiation of tumors from other lesions and the distinction between malignant and benign tumors, as these determinations significantly impact treatment decisions and therapeutic strategies. Numerous researchers have successfully developed AI algorithms based on convolutional neural networks (CNNs) for categorizing cancer whole-slide imaging (WSI) into two vital categories: carcinoma and non-carcinoma, achieving remarkable accuracies exceeding 90% [14]. Understanding the influence of stroma on tumors, Ehteshami Bejnordi et al. [15] introduced a CNN-based model that integrates stromal features to effectively differentiate invasive breast cancer from benign lesions. DL-based AI algorithms have exhibited comparable accuracy to skilled pathologists in discriminating between benign and malignant colorectal tumors [16,17], as well as in distinguishing melanoma from nevus [18]. Given these promising research outcomes, it is entirely rational to deploy AI algorithms for double-confirming high-risk slides, mitigating the risks of misdiagnosis.

However, labeling or annotating pathology slides remains a bottleneck in the development of AI-driven pathology models and applications. While weakly supervised DL models have recently alleviated the burden of labeling to some extent, annotation remains a time-consuming process. Mercan et al. [19] employed weakly supervised DL models to classify breast lesions into non-proliferative, proliferative, atypical hyperplasia, carcinoma in situ, and invasive carcinoma using WSIs of breast biopsy specimens, achieving a precision rate of 81%. Other examples include Wang et al. [20], who successfully categorized gastric lesions as normal, dysplasia, and cancer with an accuracy of 86.5%, and Tomita et al. [21], who demonstrated an accuracy of 83% in classifying esophagus lesions into Barrett esophagus, dysplasia, and cancer. In routine pathology practice, immunohistochemical staining (IHC) is a commonly used and essential ancillary diagnostic tool. The pan-cytokeratin antibody (Pan-CK) is a cocktail of all types of cytokeratin and is used for carcinoma detection because it labels all the epithelium in the slides. Therefore, pathologists often employ Pan-CK on high-risk NPC slides to aid in distinguishing reactive lymphocytes from carcinoma cells. Additional Epstein–Barr virus in situ hybridization (EBER) is used to differentiate normal mucosa epithelium from carcinoma cells.

In the current study, we aim to prove the concept that IHC-aided annotation by non-pathologists can lead to the development of a practical model for distinguishing NPC in WSIs without the time-consuming involvement of pathologists.

## 2. Materials and Methods

### 2.1. Dataset Description

In this study, we have compiled a dataset of NPC slides from 251 different patients. Each of these slides includes Pan-CK IHC slides, and in some cases, EBER slides were also included based on the original database. These slide images were collected from Cardinal Tien Hospital between 2014 and 2018, and ethical approval for their use was obtained from the medical ethics committee of Cardinal Tien Hospital. The dataset comprises diagnostic information and whole slide image (WSI) data, which include hematoxylin and eosin (H&E) slides, Pan-CK IHC slides, and EBER slides. Specifically, there are 251 H&E slides, 251 Pan-CK IHC slides, and 149 EBER slides, all obtained from 251 patients diagnosed with NPC. After data augmentation, the experimental dataset was divided into three subsets for training, testing, and validation purposes.

### 2.2. Histopathological Examination

Histopathological examination involves the microscopic study of tissue samples to assess their structure, composition, and any abnormalities. The process includes the following steps:

Fixation in formalin: Tissue samples are immersed in formalin for a minimum of 6 h. Formalin acts as a fixative, preserving the tissue’s cellular structure.

Dehydration: After fixation, the tissue samples are dehydrated, typically using a series of alcohol solutions. This process removes water from the tissues and prepares them for embedding in paraffin wax.

Embedding: The dehydrated tissue samples are embedded in paraffin blocks. Embedding in paraffin provides support and allows the samples to be thinly sectioned.

Sectioning: The paraffin-embedded tissue blocks are cut into thin sections, usually measuring 3–5 μm in thickness. These sections are placed on slides for further processing.

Staining: The thin tissue sections are stained with hematoxylin and eosin (H&E). Hematoxylin stains cell nuclei blue, and eosin stains the cytoplasm and extracellular structures pink.

### 2.3. Immunohistochemical Staining and EBV In Situ Hybridization

To perform immunohistochemical staining, tissue blocks were sectioned, and a Ventana BenchMark XT automated stainer (Ventana, Tucson, AZ, USA) was utilized. In brief, 4 μm thick consecutive sections were obtained from formalin-fixed, paraffin-embedded tissue. The primary antibody used was Pan-CK (Ventana, Tucson, AZ, USA). Additionally, EBER was conducted using the same automated stainer, with an EBV probe for detection and a primary antibody of DIG for subsequent detection. Tissue staining was visualized using a DAB substrate chromogen solution, followed by counterstaining with hematoxylin, dehydration, and mounting.

### 2.4. Data Preparation Framework

The annotation of NPC areas was carried out by a non-pathologist (TPL) using a free-hand region-of-interest tool. The annotations are enclosed shapes to the region of interest. The annotation cannot be made if the annotations are not enclosed shapes. This function is embedded in the software tool. Annotations primarily relied on the locations indicated in Pan-CK IHC images. In challenging cases, the positive EBER area confirmed the presence of NPC, while a negative EBER result excluded benign mucosa epithelium.

### 2.5. Model Development

In our patch-level training, testing, and validation procedures, we employed square image patches with dimensions of 400 × 400 pixels. These patches were randomly and dynamically cropped from the areas that were free-hand annotated. To ensure accurate representation, we had specific criteria for patch selection based on the content.

Benign tissue patches: These patches were entirely situated outside the annotated region, ensuring they solely contained benign tissue.

NPC patches: For patches representing NPC, a minimum of 90% of the patch area needed to be within the annotated tumor region.

In terms of patch distribution, we collected 6832 patches for each class, totaling 13,664 patches, with 7479 patches for benign tissue and 6832 patches for NPC.

Our training utilized ResNeXt model pretrained on ImageNet, a deep neural network architecture incorporating both residual and inception features. A training set, validation set, and test set were created by randomly splitting the samples using an 8:1:1 ratio, respectively. During training, we employed a batch size of 48 patches, consisting of 24 patches for NPC and 24 patches for benign tissue.

To ensure robustness and reliability, we conducted multiple testing inference processes, repeating them 30 times. This allowed us to compare the Receiver Operating Characteristic (ROC) curves of our original and final patch-level models and evaluate their performance consistently.

### 2.6. Computer Hardware

Considering image size, data quantity, and GPU memory, we utilized a batch size of nine images for the entire network. The entire network was implemented using TensorFlow. All experiments were carried out on an NVIDIA GeForce 1080Ti GPU (11 GB) with CUDA-9.0 and cuDNN-7.0 library optimizations.

## 3. Results

In this study, we collected slides from 251 different patients, specifically focusing on NPC. Our comprehensive dataset encompasses 251 H&E slides, 251 Pan-CK IHC slides, and 149 EBER slides. The annotation of NPC regions within the H&E slides was meticulously carried out by a non-pathologist trainee. They performed this task by cross-referencing the H&E slides with corresponding Pan-CK IHC slides, with and without EBER slides.

To ensure robustness, we performed data augmentation and divided our experimental dataset into three distinct subsets: one for training, one for testing, and another for validation. A visual representation of our study design can be found in Figure 1. Furthermore, in Figure 2, we present representative images of NPC in both H&E stain and corresponding Pan-CK immunohistochemical stain. We showcase two cases—the first being a relatively straightforward instance with NPC cells exhibiting a large pale morphology in the H&E slide (Figure 2A), prominently highlighted by the Pan-CK IHC stain (Figure 2B). The second case is more challenging, with NPC cells infiltrating the stroma in small nests, as demonstrated in the H&E slide (Figure 2C) and accentuated by the Pan-CK IHC stain (Figure 2D).

Figure 3 illustrates a complex NPC case with tumor nests primarily found in single or tiny clusters in the H&E stain (Figure 3A). This case is further illuminated by the corresponding Pan-CK IHC (Figure 3B). Lastly, we employ EBER staining (Figure 3C) to confirm the presence of NPC cells and distinguish them from normal mucosa epithelium, which typically exhibits a negative EBER result.

In Figure 4, we provide a visual representation of the non-pathologist trainee’s labeling approach. They used a free-hand region-of-interest style to annotate NPC areas, defining them by overlapping regions in the Pan-CK IHC slides, both with and without EBER slides.

Moving on to Figure 5, we examine the variation in NPC cell size across different patch dimensions, including 1000 × 1000, 400 × 400, and 200 × 200 (Figure 5A). For our patch-level training, testing, and validation, we randomly and dynamically cropped square image patches measuring 400 × 400 pixels from the previously annotated areas. In Figure 5B, we present the design of our model, which employed ResNeXt, a deep neural network incorporating a residual and inception architecture. During training, we used a batch size of 48 patches, ensuring a balanced representation of 24 patches for NPC and 24 patches for benign tissue.

Our training and test history, as depicted in Figure 6, demonstrates that after 14 epochs, our model achieved an accuracy exceeding 85% for both the training and test sets. For a comprehensive summary of our results, please refer to Table 1. Notably, the training set yielded an accuracy of 0.925, while the validation set achieved an accuracy of 0.900. In the training set, the area under the curve (AUC) for NPC was 0.945, coupled with a sensitivity of 0.985 and specificity of 0.896. In the validation set, the AUC for NPC was 0.896, with a sensitivity of 0.919 and specificity of 0.878. These results underscore the efficacy of our model in accurately identifying NPC in pathology slides.

## 4. Discussion and Conclusions

In this study, we have achieved a significant milestone by successfully applying deep convolutional neural networks to identify NPC, all without the need for expert pathologist annotations. The strategic choice of additional staining, such as Pan-CK and EBER, has streamlined the slide labeling process, making it accessible even to non-pathologist trainees. This is particularly valuable given the challenges posed by distinguishing NPC tumor cells from background inflammatory cells and normal epithelium mucosa.

Historically, NPC classification has evolved significantly since its initial characterization in 1962 by Liang et al. [22], which categorized it into three groups. Over time, the WHO has refined its pathological classification, with the current international standard being the third edition of the WHO staging (2003). Despite these advancements, current pathological classifications do not always effectively differentiate patient prognoses [23].

In the realm of NPC diagnosis, molecular indicators like peripheral blood EBV antibody and EBV DNA copy number have played a pivotal role [24,25,26,27]. These indicators have demonstrated high sensitivity and specificity, offering valuable insights for early NPC diagnosis. However, it is crucial to note that even when these molecular markers test negative, the possibility of NPC cannot be entirely ruled out. The traditional pathology diagnosis process typically involves H&E staining as the primary method. When there are cells presenting suspicious features indicative of potential malignancy, pathologists often turn to IHC markers for further clarification. Cytokeratin (CK) is a commonly used marker to distinguish epithelial cells and lymphocytes. Additionally, pathologists may employ other epithelial markers such as CK5/6 or the squamous epithelial marker p63. However, challenges arise when attempting to differentiate between benign and malignant epithelium, especially when normal epithelial cells or mucosal epithelium are present in the biopsy specimen. In such cases, the conventional epithelial or squamous epithelial markers may not provide a clear distinction. Here, the EBER emerges as a valuable marker. EBER staining is selective for malignant epithelium, making it easier for pathologists to identify and interpret on the slides. Consequently, EBER serves as an effective tool in the diagnostic process by aiding in the differentiation between benign and malignant epithelial cells.

Deep learning, a form of artificial intelligence technology, is loosely inspired by the structure and function of biological neural networks. It has shown remarkable performance in various automated image-recognition applications. Their groundbreaking work involved converting optical data into a matrix of optical density values, paving the way for computerized image analysis—a seminal moment in the inception of digital pathology. The transformation continued with the introduction of whole-slide scanners in 1999, a significant milestone that catapulted the application of AI in digital pathology. These computational approaches were tailored for the analysis of digitized whole-slide images (WSIs), opening new horizons for pathology research and diagnosis. A critical development in this journey was the establishment of extensive digital slide repositories, notably exemplified by The Cancer Genome Atlas (TCGA). These repositories provided researchers with unrestricted access to meticulously curated and annotated pathology image datasets.

Expert systems, the earliest entrants into the realm of AI in pathology, were built on predefined rules meticulously crafted by domain experts. These systems leveraged the wealth of knowledge possessed by human pathologists but were limited by their inability to adapt and generalize beyond the specific rules they were programmed with. Consequently, their applications were often constrained to well-defined, narrow domains [28].

The transition to traditional machine learning (ML) marked a significant leap in the field. In this phase, pathologists and AI researchers began to define features based on ex-pert knowledge. ML algorithms, such as decision trees, support vector machines, and random forests, learned from these feature-engineered datasets to make predictions. This approach offered more flexibility compared to expert systems, as it allowed for the integration of a broader range of data and patterns. However, the need for manual feature engineering remained a bottleneck, and the performance was heavily reliant on the quality of these handcrafted features.

Recently, several deep learning-based algorithms have been created for the detection of cancer and other applications in the field of pathology [28,29,30]. Initial findings indicate that some of these algorithms can even outperform human pathologists in terms of sensitivity when it comes to detecting individual cancer foci in digital images. Sensitivity refers to the ability to correctly identify true positive cases, which, in this context, means accurately detecting cancer. These algorithms are highly sensitive and proficient at identifying cancerous regions [31,32].

However, there is a trade-off. Their heightened sensitivity often comes at the cost of increased false positives, which are instances where the algorithm incorrectly identifies non-cancerous regions as cancer. This increase in false positives can potentially limit the practical utility of these algorithms for automated clinical use, as it may lead to unnecessary concerns or interventions. Striking a balance between sensitivity and specificity is a critical challenge in the development and application of these algorithms in pathology [31].

Computer algorithms have the potential to significantly enhance the workflow of pathologists. However, their widespread adoption in clinical practice is still a work in progress, and they continue to undergo regulatory scrutiny. Despite achieving favorable performance metrics, questions about the safety and quality of these algorithms persist. Additionally, many practicing pathologists may not possess an in-depth technical understanding of these algorithms, including their diagnostic accuracy, error rates, and practical utility in clinical settings.

The regulatory approval and gradual integration of whole-slide scanners have paved the way for the digitization of glass slides, enabling remote consultations and archival purposes. However, the process of digitization alone may not automatically improve the consistency or efficiency of a pathologist’s primary workflow. In some cases, digital image review might even be slightly slower than traditional glass slide review, especially for pathologists with limited experience in digital pathology.

Nonetheless, the field of digital pathology and image analysis tools holds promise in various areas. For example, it can reduce inter-reader variability in the assessment of breast cancer HER2 status [33,34]. Moreover, the digitization of pathology slides creates opportunities for the development and integration of AI-based assistive tools. These tools have the potential to enhance workflow efficiency, improve consistency, mitigate fatigue, and increase diagnostic accuracy, representing a significant advancement in the field of pathology.

Our study demonstrates that Pan-CK and EBER results can significantly aid in distinguishing NPC from benign tissue, even when conducted by non-pathologists. The secondary achievement of our model lies in its ability to reduce reliance on the pathologist’s expertise. Our model exhibits diagnostic proficiency comparable to senior pathologists, making it a valuable second-opinion tool. This is especially important as pathologists may experience reduced diagnostic accuracy when dealing with a high caseload.

IHCs serve as valuable supplements to H&E histology in pathology. In the present study, it is demonstrated that the careful selection of appropriate IHCs or other stain markers, such as EBER, can overcome the rate-limiting step of annotation performed by pathologists. Annotation, being a time-consuming process, has posed a bottleneck that hinders the swift development of AI pathology.

Pathology, with its myriad of different diagnoses encountered in daily practice, presents a diverse and complex landscape. The urgency for faster development of AI pathology models is underscored by the pressing need for these models to provide real assistance in day-to-day pathology work. By leveraging the complementary information provided by IHCs or specific stain markers, researchers and pathologists aim to streamline the annotation process, ultimately expediting the development and deployment of AI models in pathology for more efficient and accurate diagnoses.

The integration of artificial intelligence into pathology has created a growing demand for high-quality pathological image datasets. The process of pathological annotation is traditionally time-consuming and relies on professional pathologists, posing a bottleneck for AI model development. Our study highlights the importance of using AI to alleviate this burden and enhance diagnostic capabilities.

Indeed, the use of computer-assisted diagnosis, particularly leveraging technologies such as deep learning, is widely acknowledged for its potential to enhance the capabilities of pathologists [35]. One of the practical applications at the slide level involves the automated identification of negative or challenging slides for IHC staining before the pathologist’s review, with the aim of streamlining the pathologist’s workflow. When applied in this manner, all negative cases could be verified through upfront IHC staining. However, it is important to note that this comes at an additional cost of three slides in both datasets studied. Alternatively, the algorithm could serve to prioritize the review of positive cases, expediting the sign-out process for cases with positive results. Another potential use case involves a “second read” to flag missed malignancy for review, particularly as part of an institutional or individual Quality Assurance protocol. Finally, at the patch level, an assisted read mode could guide pathologists to highly suspicious regions, akin to the role of a junior resident marking regions of interest (ROIs) with ink on a physical glass slide.

Digital assistance can be integrated into clinical workflows in other ways. It could serve as a “screening” tool to differentiate clearly negative and/or positive cases, akin to the FDA-approved use of computer assistance for cervical cytology specimens [36]. Alternatively, it could function as a “second read” for challenging cases following the primary pathologist’s review. The current model can be served a tool that eases the review burden for negative cases or triages complex cases for IHC before the initial pathologist review has the potential to reduce reporting delays and offer cost savings through efficiency and improved accuracy. While the potential benefits of these approaches are promising, careful consideration of their limitations, along with clear instructions and use-case definitions, will be crucial for the successful and safe integration of assistive tools in the field of pathology.

However, there are limitations to our study. It was retrospective in nature, and the data had some deficiencies in terms of completeness and homogeneity. Additionally, the interpretability of deep learning algorithms remains a challenge. The results of the precision rate in our study are reported to be lower than those observed in another study [35]. However, the primary objective of the current study is to establish the concept that non-pathologist annotation can still contribute to the development of a robust AI model capable of distinguishing NPC in histological slides. It is emphasized that the study’s focus is on proving the feasibility and potential of leveraging non-pathologist annotations.

The acknowledgment that the precision rate is lower in the current study is accompanied by the assertion that increasing the number of cases could lead to improved results. The suggestion is that with a larger dataset, the precision rate could potentially align more closely with the outcomes reported in other studies. This implies that, despite the current limitations, the fundamental concept and methodology presented in the study lay the groundwork for future investigations with expanded datasets to further refine and validate the AI model’s performance. Despite these limitations, our results are promising, demonstrating the potential of laboratory techniques to reduce the workload of pathologists.

In conclusion, our study underscores the utility of IHC and strategic marker selection in facilitating slide annotation, even by non-pathologists. This approach has the potential to significantly reduce the workload of pathologists, accelerate the development of AI models in pathology, and improve cancer detection in the future.

## Figures and Tables

**Figure 1 diagnostics-13-03685-f001:**
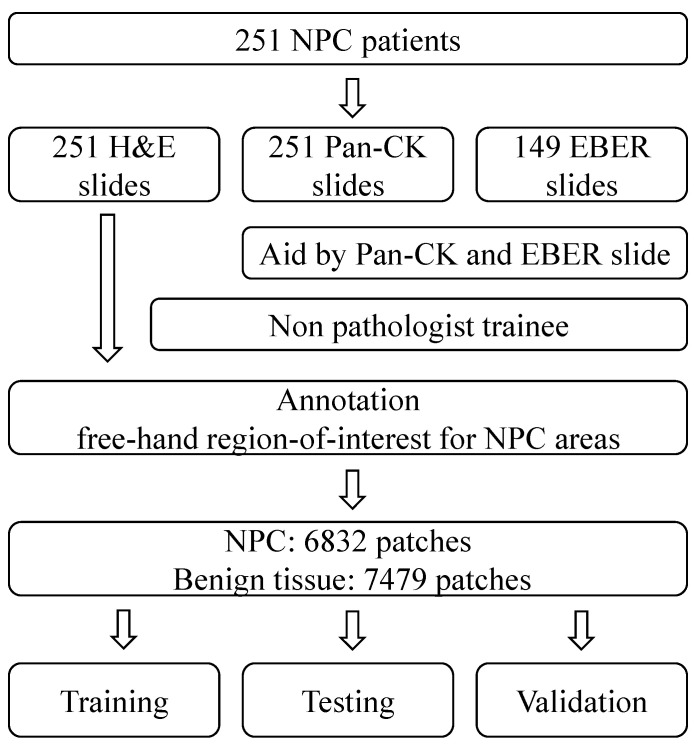
The study design of the current study. NPC, nasopharyngeal carcinoma; Pan-CK, pan-cytokeratin antibody; EBER, Epstein–Barr virus in situ hybridization.

**Figure 2 diagnostics-13-03685-f002:**
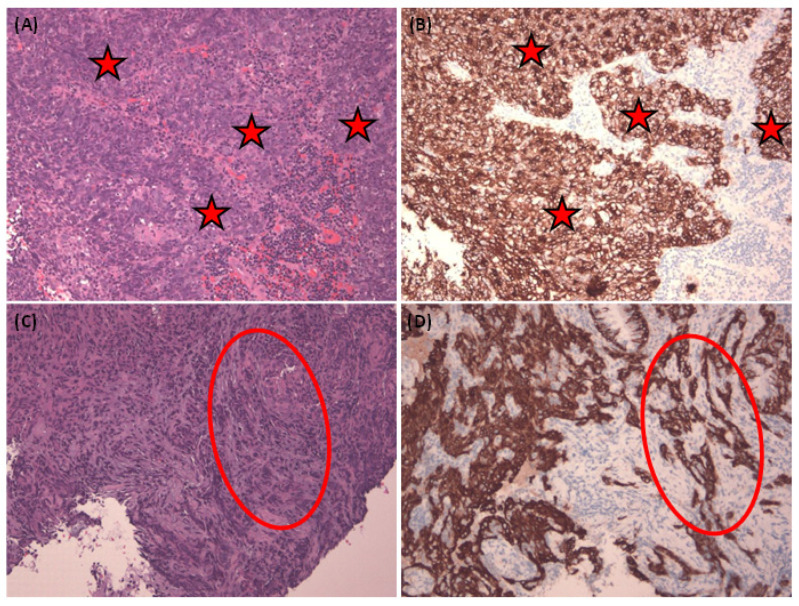
Representative pictures of nasopharyngeal carcinoma and corresponding Pan-CK immunohistochemical stain. Representative pictures of nasopharyngeal carcinoma in H&E stain (**A**,**C**) and corresponding Pan-CK immunohistochemical stain (**B**,**D**). The easier case with NPC cells (red asterisks) in a large pale morphology in H&E slide (**A**) and highlighted by Pan-CK IHC stain (**B**). Harder NPC cases with NPC cells infiltrating (red circle) in the stroma in a small nest showed in the H&E slide (**C**) and also highlighted by Pan-CK IHC stain (**D**). All figures are in 100× magnification.

**Figure 3 diagnostics-13-03685-f003:**
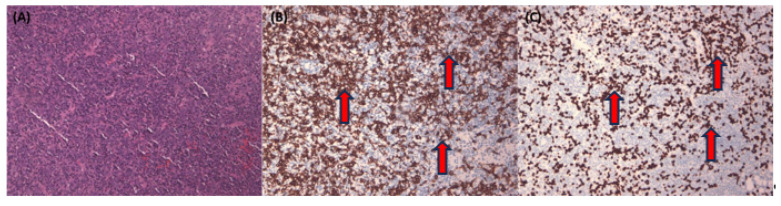
Representative pictures of difficult nasopharyngeal carcinoma case with H&E stain and corresponding Pan-CK immunohistochemical stain and EBER stain. Representative pictures of difficult nasopharyngeal carcinoma cases, with mostly single or tiny tumor nests (red arrows) in H&E stain (**A**) and corresponding Pan-CK IHC (**B**) and EBER stain (**C**). All figures are in 100× magnification.

**Figure 4 diagnostics-13-03685-f004:**
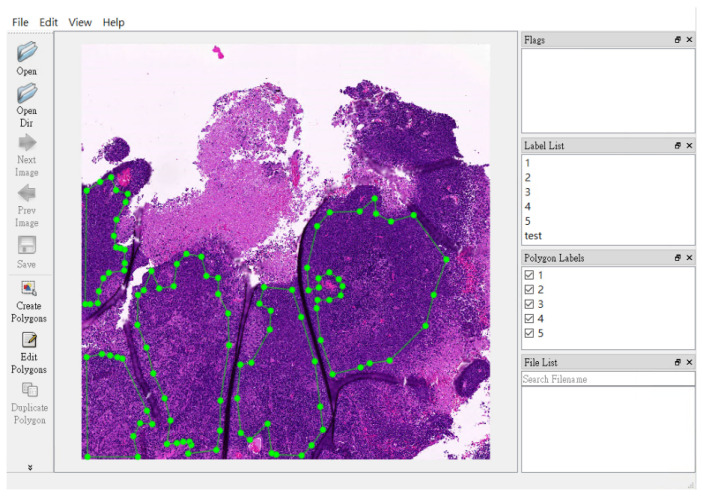
Representative pictures of free-hand region-of-interest labelling in nasopharyngeal carcinoma slides. The non-pathologist trainee label the NPC area in a free-hand region-of-interest style for NPC areas (green dots area). They labelled the NPC area by simultaneous the area of pan-CK IHC slides with/without EBER slides.

**Figure 5 diagnostics-13-03685-f005:**
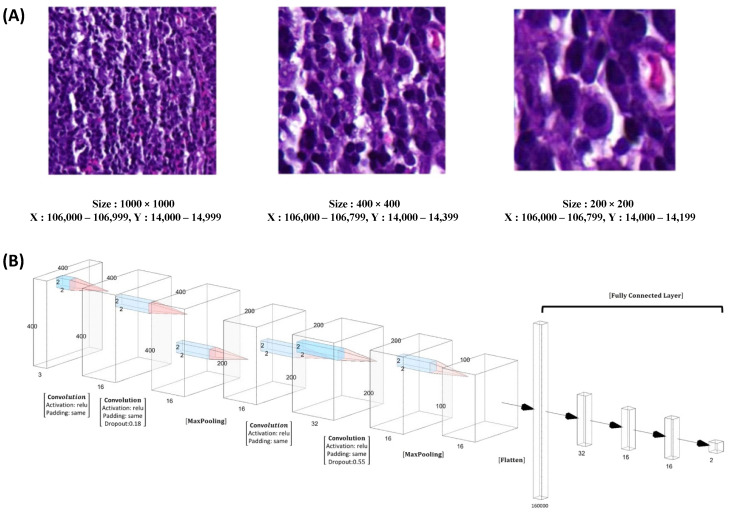
The representative pictures of different size of the patch and model design. The representative NPC cells size in different patch size of 1000 × 1000, 400 × 400, and 200 × 200 (**A**). In our patch-level training, testing, and validation procedures, we employed square image patches with dimensions of 400 × 400 pixels. These patches were randomly and dynamically cropped from the areas that were free-hand annotated. (**B**). Our training utilized ResNeXt, a deep neural network architecture incorporating both residual and inception features. During training, we employed a batch size of 48 patches, consisting of 24 patches for NPC and 24 patches for benign tissue.

**Figure 6 diagnostics-13-03685-f006:**
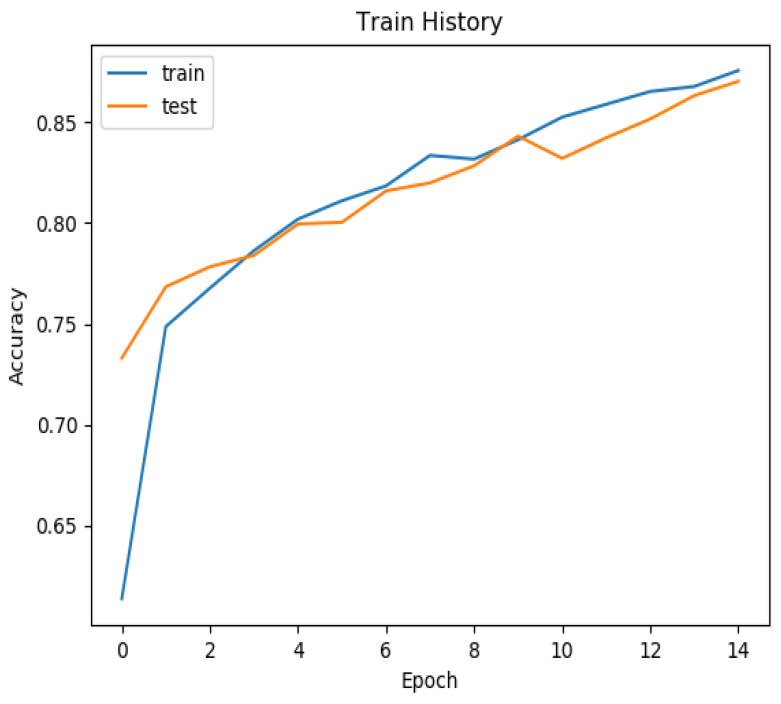
The training history of the train set and test set. The training history of the accuracy and epochs. After 14 epochs, the accuracy reached more than 85% in both the train and test set.

**Table 1 diagnostics-13-03685-t001:** Result of training and validation sets of the model.

Evaluation Indicator	Accuracy	Category	AUC	Sensitivity	Specificity
Training set	0.925	NPC	0.945	0.985	0.896
		Benign tissue	0.905	0.986	0.878
Validation set	0.900	NPC	0.896	0.919	0.878
		Benign tissue	0.904	0.920	0.869

AUC, area under curve; NPC, nasopharyngeal carcinoma.

## Data Availability

The dataset supporting the conclusions of this article is included with the article.

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
