# Peer review of "Immunohistochemical Stain-Aided Annotation Accelerates Machine Learning and Deep Learning Model Development in the Pathologic Diagnosis of Nasopharyngeal Carcinoma"

_diagnostics, 2023, doi:10.3390/diagnostics13243685_

Round 1

Reviewer 1 Report

Comments and Suggestions for Authors

line 88, cite a reference to back up your claim.

line 99, same as above. list a few publications of the applications shown here.

line 268. how to ensure annotations are enclosed shapes?

line 281. how was this 1:1 split was guaranteed in each batch? isn't data were randomly selected from training set?

line 284, did you fine tune the Resnet network model? if so, please provide some additional details?

also, can you elaborate on the reason for using IHC and EBER stained slides to assist H&E stained slide annotations? why not use IHC directly. Some background information about choice of stains would be helpful

Comments on the Quality of English Language

Overall, the paper is well-written. 

Author Response

Dear reviewers:

First, we want to thank you for your suggestions since they are very helpful both for the present manuscript and for our future paper-writing. We answered correspondingly to all the suggestions and questions from the reviewers. The followings are our answers to the questions:

Comment 1-1:

line 88, cite a reference to back up your claim.
 Authors’ responses:

Thank reviewer’s excellent comment. We add the reference in this claim.

Ahmad Z, Rahim S, Zubair M, Abdul-Ghafar J. Artificial intelligence (AI) in medicine, current applications and future role with special emphasis on its potential and promise in pathology: present and future impact, obstacles including costs and acceptance among pathologists, practical and philosophical considerations. A comprehensive review. Diagn Pathol. 2021 Mar 17;16(1):24.

Comment 1-2:

line 99, same as above. list a few publications of the applications shown here. 

Authors’ responses:

Thank reviewer’s excellent comment. We add some reference in this claim.

  1. Bera K, Schalper KA, Rimm DL, Velcheti V, Madabhushi A. Artificial intelligence in digital pathology - new tools for diagnosis and precision oncology. Nat Rev Clin Oncol. 2019 Nov;16(11):703-715.
  2. Parwani AV, Patel A, Zhou M, Cheville JC, Tizhoosh H, Humphrey P, Reuter VE, True LD. An update on computational pathology tools for genitourinary pathology practice: A review paper from the Genitourinary Pathology Society (GUPS). J Pathol Inform. 2022 Dec 30;14:100177.
  3. Plass M, Kargl M, Kiehl TR, Regitnig P, Geißler C, Evans T, Zerbe N, Carvalho R, Holzinger A, Müller H. Explainability and causability in digital pathology. J Pathol Clin Res. 2023 Jul;9(4):251-260.

Comment 1-3:

line 268. how to ensure annotations are enclosed shapes?
Authors’ responses:

        Thank reviewer’s excellent comment. The annotation cannot be made if the annotations are not enclosed shapes. This function is the embedded in the software tool. In addition, we add the sentences in the line 161-162.

Comment 1-4:

line 281. how was this 1:1 split was guaranteed in each batch? isn't data were randomly selected from training set?

Authors’ responses:

Thank reviewer’s excellent comment. A training set, validation set, and test set were created by randomly splitting the samples using an 8:1:1 ratio, respectively. We also add the sentence in the line 177-179.

Comment 1-5:

line 284, did you fine tune the Resnet network model? if so, please provide some additional details?

Authors’ responses:

Thank reviewer’s excellent comment. Our training utilized ResNeXt model pretrained on ImageNet, a deep neural network architecture incorporating both residual and inception features. A training set, validation set, and test set were created by randomly splitting the samples using a 8:1:1 ratio, respectively. During training, we employed a batch size of 48 patches, consisting of 24 patches for NPC and 24 patches for benign tissue. We also add the sentences in the line 176-180.

Comment 1-6:

Can you elaborate on the reason for using IHC and EBER stained slides to assist H&E stained slide annotations? why not use IHC directly. Some background information about choice of stains would be helpful.

Authors’ responses:

Thank reviewer’s excellent comment. The traditional pathology diagnosis process typically involves H&E staining as the primary method. When there are cells presenting suspicious features indicative of potential malignancy, pathologists often turn to IHC markers for further clarification. Cytokeratin (CK) is a commonly used marker to distinguish epithelial cells and lymphocytes. Additionally, pathologists may employ other epithelial markers such as CK5/6 or the squamous epithelial marker p63. However, challenges arise when attempting to differentiate between benign and malignant epithelium, especially when normal epithelial cells or mucosal epithelium are present in the biopsy specimen. In such cases, the conventional epithelial or squamous epithelial markers may not provide a clear distinction. Here, the EBER emerges as a valuable marker. EBER staining is selective for malignant epithelium, making it easier for pathologists to identify and interpret on the slides. Consequently, EBER serves as an effective tool in the diagnostic process by aiding in the differentiation between benign and malignant epithelial cells. We also add the sentences in the line 344-356.

Thank you again for giving us this opportunity to revise our manuscript so that our work is improved.

Sincerely

Yen-Lin CHEN, M.D Ph.D.

Reviewer 2 Report

Comments and Suggestions for Authors

- The Introduction should be shorter.

- NPC, IHC abbreviations are repeated.

- How is NPC diagnosed pathologically in clinical settings without AI? How is it different from your set? Any other IHC markers?

- Can you mark the narrated specific features in the pictures with arrows?

- What is the clinical significance of this study? How can it be used in clinical settings? As a second read?

- How is your results (the precision rate) different from other studies reported in the literature?

- Lastly, isn’t it evident that immunohistochemical stain supplements H & E histology?

Comments on the Quality of English Language

 Minor editing of English language required.

Author Response

Dear reviewers:

First, we want to thank you for your suggestions since they are very helpful both for the present manuscript and for our future paper-writing. We answered correspondingly to all the suggestions and questions from the reviewers. The followings are our answers to the questions:

Comment 2-1:

The Introduction should be shorter.

Authors’ responses:

Thank reviewer’s excellent comment. We shortened the introduction.

Comment 2-2:

NPC, IHC abbreviations are repeated.

Authors’ responses:

Thank reviewer’s excellent comment. We revised the repeated abbreviation.

Comment 2-3:

How is NPC diagnosed pathologically in clinical settings without AI? How is it different from your set? Any other IHC markers?

Authors’ responses:

Thank reviewer’s excellent comment. The traditional pathology diagnosis process relies primarily on H&E staining without the assistance of AI. Occasionally, IHCs are employed to highlight specific cellular features, such as epithelial cells and cancer cells. In EBER stain, is utilized to differentiate between normal epithelial cells and cancer cells. In current clinical pathology practice, pathologists serve as the sole diagnostic decision-makers without the aid of AI.

However, the pathologists may face challenges such as fatigue or the potential to overlook malignancies, particularly when dealing with a large volume of slides. The introduction of AI is envisioned as a supportive measure, offering a second look to assist pathologists in their initial assessment. The AI's role is characterized as a secondary review, providing an additional layer of scrutiny to help ensure thorough and accurate evaluations. As pathologists gain more experience in interpreting AI results, there is a contemplation of transitioning AI from a second-line review to a front-line screening tool. However, the acknowledgment is made that the field is currently in the early stages of AI development in pathology, and more studies are deemed necessary for further exploration and validation of AI's capabilities in the diagnostic process.

Comment 2-4:

Can you mark the narrated specific features in the pictures with arrows?

Authors’ responses:

Thank reviewer’s excellent comment. We add some markers in the pictures.

Comment 2-5:

What is the clinical significance of this study? How can it be used in clinical settings? As a second read?

Authors’ responses:

Thank reviewer’s excellent comment. Our study demonstrates that Pan-CK and EBER results can significantly aid in distinguishing NPC from benign tissue, even when conducted by non-pathologists. The secondary achievement of our model lies in its ability to reduce the reliance on the pathologist's expertise. Our model exhibits diagnostic proficiency comparable to senior pathologists, making it a valuable second opinion tool. This is especially important as pathologists may experience reduced diagnostic accuracy when dealing with a high caseload.

Comment 2-6:

How is your results (the precision rate) different from other studies reported in the literature?

Authors’ responses:

Thank reviewer’s excellent comment. The results of the precision rate in our study are reported to be lower than those observed in other studies. However, the primary objective of the current study is to establish the concept that non-pathologist annotation can still contribute to the development of a robust AI model capable of distinguishing NPC in histological slides. It is emphasized that the study's focus is on proving the feasibility and potential of leveraging non-pathologist annotations.

The acknowledgment that the precision rate is lower in the current study is accompanied by the assertion that increasing the number of cases could lead to improved results. The suggestion is that with a larger dataset, the precision rate could potentially align more closely with the outcomes reported in other studies. This implies that, despite the current limitations, the fundamental concept and methodology presented in the study lay the groundwork for future investigations with expanded datasets to further refine and validate the AI model's performance. We also add the sentences in the line 463-475.

Comment 2-7:

Lastly, isn’t it evident that immunohistochemical stain supplements H & E histology?

Authors’ responses:

Thank reviewer’s excellent comment. IHCs serve as valuable supplements to H&E histology in pathology. In the present study, it is demonstrated that the careful selection of appropriate IHCs or other stain markers, such as EBER, can overcome the rate-limiting step of annotation performed by pathologists. Annotation, being a time-consuming process, has posed a bottleneck that hinders the swift development of AI pathology.

Pathology, with its myriad of different diagnoses encountered in daily practice, presents a diverse and complex landscape. The urgency for faster development of AI pathology models is underscored by the pressing need for these models to provide real assistance in day-to-day pathology work. By leveraging the complementary information provided by IHCs or specific stain markers, researchers and pathologists aim to streamline the annotation process, ultimately expediting the development and deployment of AI models in pathology for more efficient and accurate diagnoses. We also add the sentences in the line 419-430.

Comment 2-8:

Minor editing of English language required.

Authors’ responses:

Thank reviewer’s excellent comment. We did the English revise.

Thank you again for giving us this opportunity to revise our manuscript so that our work is improved.

Sincerely

Yen-Lin CHEN, M.D Ph.D.